# DFSAttn: Dynamic Fine-grained Sparse Attention for Efficient Video Generation

Jie Hu[1]  Zixiang Gao[1]  Yutong He[1]  Kun Yuan[1]

## Abstract

Diffusion transformers have achieved remarkable success in high-quality video generation, yet their reliance on spatiotemporal 3D full attention incurs prohibitive computational cost due to the quadratic complexity of attention. Block sparse attention is a common approach to mitigate this by focusing computation on important regions. However, attention maps in DiTs exhibit inherently dynamic and fine-grained sparsity, which causes existing block sparse attention methods to degrade significantly in quality, especially at high sparsity ratios. In this paper, we revisit block sparse attention and derive a theoretical lower bound on attention recall to characterize the key factors governing its effectiveness. Guided by these insights, we propose DFSAttn, a training-free sparse attention framework that enables dynamic, fine-grained sparsification efficiently. DFSAttn incorporates three core designs: Hilbert curve–based token reordering to achieve fine-grained sparsity while preserving efficient GPU execution, hierarchical block scoring for accurate block importance estimation, and sparse mask caching with adaptive ratios to balance accuracy and efficiency. Experimental results demonstrate that DFSAttn consistently outperforms prior methods under high sparsity, achieving up to $2.1\times$ end-to-end speedup while maintaining high generation quality. Our code is open-sourced and available at this link.

## 1. Introduction

Diffusion Transformers (DiTs) (Peebles & Xie, 2023) have achieved remarkable success in various applications. Recent state-of-the-art video DiTs (Yang et al., 2025; Zheng et al., 2024; Kong et al., 2024; Wan et al., 2025) demonstrate impressive capability in synthesizing high-fidelity videos by adopting spatiotemporal 3D full attention. Despite its effectiveness, 3D full attention incurs prohibitive computational overhead due to the quadratic computational complexity of attention (Vaswani et al., 2017). For example, generating a 129-frame 720p video with HunyuanVideo (Kong et al., 2024) requires approximately 30 minutes on an NVIDIA H100 PCIe GPU, severely limiting practical deployment.

Fortunately, attention maps exhibit inherent sparsity (Zhang et al., 2023; Xiao et al., 2024), where only a small subset of critical tokens dominate the output. Sparse attention methods exploit this property by constructing sparse masks to omit redundant computations. In particular, block sparse attention, which skips computations at the block level, aligns naturally with modern hardware and efficient kernels such as FlashAttention (Dao et al., 2022), enabling high-performance attention computation.

Existing training-free sparse attention methods for DiTs can be broadly categorized into static and dynamic approaches, depending on whether the block sparse mask is fixed offline or constructed on-the-fly during inference. Static methods (Yuan et al., 2024; Xi et al., 2025; Zhang et al., 2025b; Chen et al., 2026; Zhao et al., 2026; Li et al., 2026) rely on empirically designed mask patterns, while dynamic methods (Zhang et al., 2025a; Xu et al., 2025; Zhang et al., 2026; Xia et al., 2025; Shen et al., 2025; Yang et al., 2026; Liu et al., 2026a;b) aim to identify important blocks during inference. However, methods from both paradigms suffer from severe quality degradation under high sparsity. We argue that this limitation stems from a fundamental mismatch between GPU-efficient block-wise attention operations and the distinctive attention characteristics of DiTs, which are not adequately captured by existing designs.

Our first key observation is that attention maps in DiTs exhibit a dynamic and fine-grained sparsity pattern. As illustrated in Figure 2, attention patterns vary significantly across layers and heads, with salient interactions scattered throughout the attention map. Consequently, static or coarse-grained sparse masks inevitably discard critical dependencies and lead to accuracy loss. Moreover, we observe that block sparse attention becomes progressively more effective as the diffusion process advances (Figure 3), indicating that the effectiveness of sparsification is closely tied to the evolving statistical structure of the attention score matrix.

[1]Peking University. Correspondence to: Kun Yuan <kunyuan@pku.edu.cn>.

*Proceedings of the 43rd International Conference on Machine Learning*, Seoul, South Korea. PMLR 306, 2026. Copyright 2026 by the author(s).

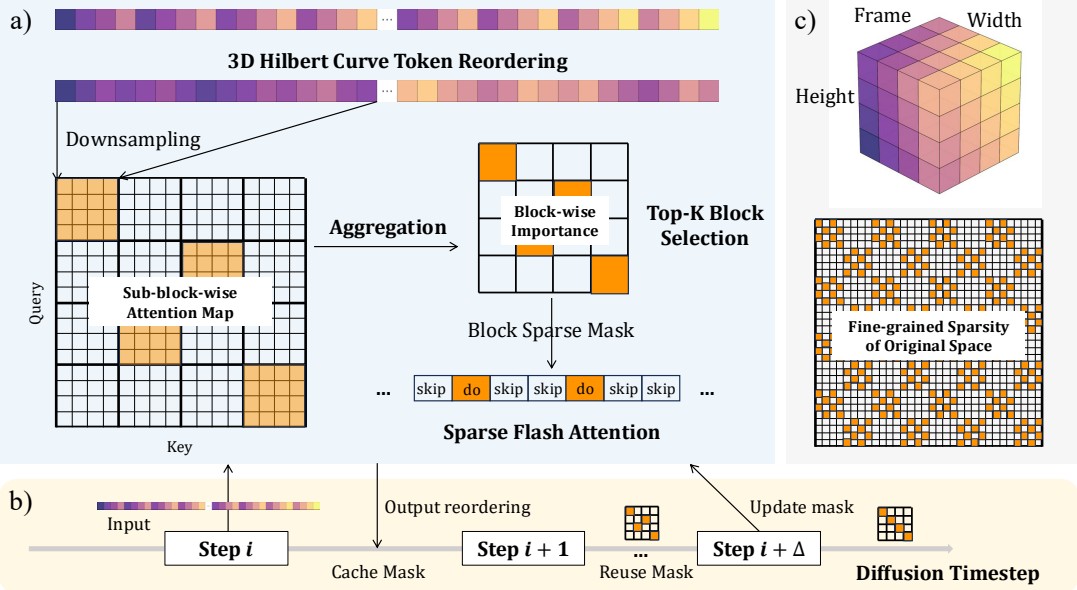

*Figure 1.* Overview of DFSAttn: a) Video tokens are reordered using a 3D Hilbert curve, and sub-block attention scores are aggregated to estimate block-wise importance. The resulting masks are applied via sparse Flash Attention, while cross-attention remains dense to preserve text–video alignment. b) Sparse masks are cached and reused across diffusion timesteps. c) Structured block sparsity in the reordered sequence induces fine-grained sparsity of original spatiotemporal space.

To systematically investigate the underlying factors of effective block sparse attention, we develop a token representation model and derive a theoretical lower bound on attention recall, which characterizes the fraction of important attention interactions preserved after sparsification. Our analysis identifies three key factors governing sparsification quality: the sparsity budget, the inter-block similarity gap, and the block-level semantic diversity.

Motivated by these insights, we introduce DFSAttn, a training-free dynamic fine-grained sparse attention method for accelerating video generation. DFSAttn incorporates three core designs, each directly addressing one of the identified factors. First, DFSAttn reorders video tokens using a 3D Hilbert curve, implicitly inducing fine-grained sparsity while aligning with GPU execution. This reordering preserves spatiotemporal locality in the 1D token sequence, thereby enlarging the inter-block similarity gap. Second, DFSAttn introduces a hierarchical block scoring mechanism that refines block importance estimation through sub-block aggregation, alleviating inaccuracies caused by mixed semantics within a block. Finally, DFSAttn adaptively reallocates the sparsity budget across diffusion steps and employs sparse mask caching, enabling a favorable balance between generation quality and efficiency.

We evaluate DFSAttn on state-of-the-art video diffusion models: HunyuanVideo (Kong et al., 2024), and Wan2.1 (Wan et al., 2025). Experimental results show that DFSAttn

consistently outperforms existing work in generation quality under high sparsity. Specifically, DFSAttn achieves an end-to-end speedup of $2.1\times$ at an 80% sparsity ratio on HunyuanVideo, while maintaining high visual fidelity, reaching PSNR up to 29.38. Our contributions are as follows:

- We identify the mismatch between block-wise sparse attention and DiTs' distinctive attention patterns, and derive a theoretical lower bound on attention recall to characterize key factors governing effective block sparse attention.

- We introduce DFSAttn, a training-free sparse attention framework that enables dynamic and fine-grained sparsification via token reordering, hierarchical block scoring, and adaptive sparse mask caching.

- DFSAttn significantly improves video generation quality under high sparsity and delivers substantial end-to-end speedups across video diffusion models.

## 2. Related Work

### 2.1. Efficient video generation

Numerous techniques have been developed to accelerate diffusion models. One major direction reduces inference cost by decreasing the number of sampling steps, through improved solvers or schedules such as DDIM (Song et al., 2020) and DPM-Solver (Lu et al., 2022), or via distilla-

tion and consistency-based models (Salimans & Ho, 2022; Song et al., 2023) that approximate the full diffusion trajectory with few steps. System-level approaches, such as DistriFusion (Li et al., 2024), parallelize diffusion inference across multiple devices. Another line of work exploits inter-timestep redundancy by caching and reusing intermediate activations or attention results, including PAB (Zhao et al., 2025), TeaCache (Liu et al., 2025), and AdaCache (Kahatapitiya et al., 2025). These caching-based approaches reduce redundant computation across diffusion steps but do not modify the attention computation within each step. DFSAttn is orthogonal to these methods and can be combined with them for further acceleration.

## 2.2. Sparse attention

Sparse attention has been widely studied in large language models to alleviate the quadratic complexity of attention (Zhang et al., 2023; Xiao et al., 2024; Jiang et al., 2024; Xu et al., 2025; Lai et al., 2025; Zhang et al., 2025a). More recently, sparse attention mechanisms have been adapted to DiTs for video generation, typically operating at the block level for GPU efficiency. Existing methods can be broadly categorized into static and dynamic approaches. Static methods predefine sparsity patterns offline, such as spatial-temporal masks (Xi et al., 2025), 3D sliding windows (Zhang et al., 2025b), energy-decay-based radial sparsity (Li et al., 2026). While efficient, these fixed patterns lack flexibility. Dynamic methods identify sparse masks on-the-fly during inference, aiming to select critical blocks. XAttention (Xu et al., 2025) estimates block importance using the sum of antidiagonal values of attention scores, and SpargeAttn (Zhang et al., 2025a) employs a two-stage online filter based on block-wise mean values. SVG2 (Yang et al., 2026) applies k-means clustering to tokens and leverages cluster centroids for selection. MOD-DiT (Liu et al., 2026b) predicts dynamic sparse masks by modeling the mixture of attention distributions across denoising steps. However, existing methods suffer from severe quality degradation under high sparsity, as they rely on coarse block-level representations that do not fully capture the fine-grained sparsity patterns. Recent work FG-Attn (Durvasula et al., 2025) proposes a fine-grained sparse attention kernel for DiTs. In contrast, DFSAttn is a kernel-free approach that leverages fine-grained sparsity through token reordering and block-wise execution.

# 3. Preliminary

## 3.1. 3D full attention in DiTs

State-of-the-art video diffusion transformers (Kong et al., 2024; Wan et al., 2025) process videos by first encoding a 3D video clip into a latent representation using a VAE. The resulting latent tensor has spatial–temporal dimensions

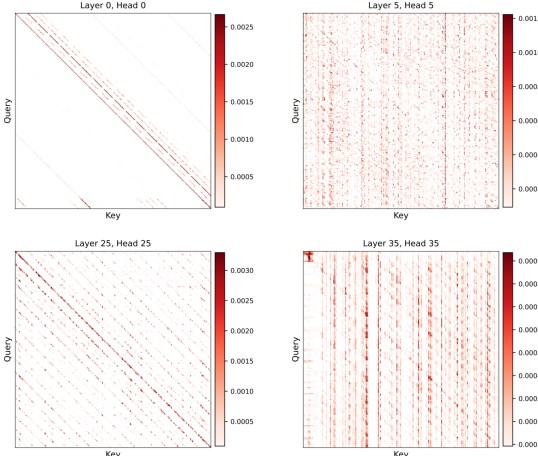

*Figure 2.* 3D full attention maps in DiTs exhibit dynamic and fine-grained sparsity patterns.

$(f, h, w)$, corresponding to the number of frames, height, and width, respectively. This 3D latent is then flattened into a single token sequence before being fed into the transformer. Consequently, the input sequence length for each transformer block is $N = f \times h \times w$. For simplicity, we omit text-conditioning tokens here, as their length is typically negligible compared to video tokens.

Within each transformer block, DiTs employ 3D full attention, where each video token attends to all others across all dimensions. Formally, for a single attention head, let $Q, K, V \in \mathbb{R}^{N \times d}$ denote the query, key, and value, where $d$ is the head dimension. Attention is computed as

$$A = \mathrm{softmax}\left(\frac{QK^\top}{\sqrt{d}}\right), \quad O = AV, \qquad (1)$$

where $A \in \mathbb{R}^{N \times N}$ is the attention score matrix and $O$ is the output. While 3D full attention enables rich spatiotemporal interactions, its computational complexity scales quadratically with $N$. As a result, attention becomes the dominant bottleneck in video generation, especially for high-resolution or long-duration videos where $N$ is large.

## 3.2. Block sparse attention

Sparse attention methods reduce computational cost by applying a binary mask $\mathcal{M} \in \{0, 1\}^{N \times N}$ to the attention matrix, yielding $\widetilde{A} = A \odot \mathcal{M}$. However, such element-wise sparsity is poorly aligned with the execution of modern GPU attention kernels. Efficient implementations such as FlashAttention (Dao et al., 2022) compute attention in a block-wise manner to reduce memory overhead. As a result, element-wise sparse attention often fails to deliver practical speedups.

Block sparse attention addresses this by partitioning the

sequence into $M = N/B$ blocks of size $B$ (with $N$ padded to be divisible by $B$), and applying sparsity via a block mask $\mathcal{M} \in \{0,1\}^{M \times M}$. Dynamic block sparse attention methods construct this mask by estimating block importance without explicitly computing the full attention matrix. Prior approaches typically compute a block-wise attention score matrix $\hat{A}$ and derive $\mathcal{M}$ by ranking these scores. Specifically, queries and keys are grouped into blocks and form block-level representations $\hat{Q} = [\hat{q}_1, \ldots, \hat{q}_M]$ and $\hat{K} = [\hat{k}_1, \ldots, \hat{k}_M]$ using a method-dependent function. The block-wise attention score is then computed as

$$\hat{A} = \text{softmax}\left(\frac{\hat{Q}\hat{K}^\top}{\sqrt{d}}\right). \quad (2)$$

The matrix $\hat{A}$ serves as a proxy for the importance of interactions between query and key blocks. For each query block, only the key blocks with the highest scores in $\hat{A}$ are retained. To quantify sparsification quality, we measure the attention score recall

$$\mathcal{R} = \frac{\|\widetilde{A}\|_1}{\|A\|_1}, \quad (3)$$

which captures the fraction of attention mass preserved after applying the sparse mask.

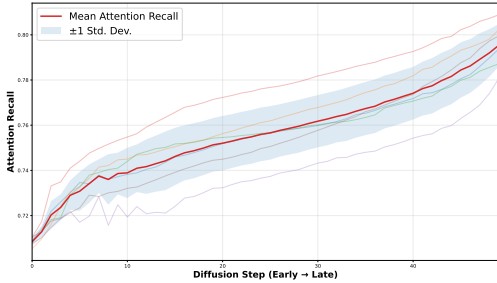

*Figure 3.* The mean attention recall (solid line) rises monotonically across diffusion steps, with low variance across various samples (shaded region).

## 4. Motivation

In this section, we identify the intrinsic sparsity patterns of attention in diffusion transformers and show that the effectiveness of block sparse attention is fundamentally governed by the statistical structure of attention scores. To formalize this connection, we introduce a token representation model and derive a theoretical lower bound on attention recall under block sparsity. Our analysis reveals three key factors that determine performance: the sparsity budget, the inter-block similarity gap, and the block-level semantic diversity.

### 4.1. Experimental observation

To better exploit attention sparsity in DiTs, we begin with an empirical analysis of attention maps across layers and attention heads. As illustrated in Figure 2, sparsity patterns vary substantially both across layers and among heads, indicating that static sparsity schemes may be suboptimal. This observation motivates the design of dynamic sparsity mechanisms that can adapt to various patterns.

We further study a typical block sparse attention scheme that constructs block-level representations via block-wise mean pooling and evaluate its attention score recall. As shown in Figure 3, the recall, averaged across all layers and attention heads, consistently increases as the diffusion process proceeds. Notably, this improvement occurs with a fixed sparsity ratio, suggesting that block sparse attention becomes more effective at later denoising steps.

This trend can be attributed to the inherent dynamics of diffusion models. At early denoising steps, latent representations are dominated by noise, leading to diffuse attention distributions. As denoising proceeds, the latents gradually converge toward the data manifold, resulting in increasingly concentrated attention. Motivated by these observations, we develop a probabilistic model of attention in the following section to systematically analyze the factors governing the performance of block sparse attention.

### 4.2. Theoretical lower bound

We develop a token representation model to characterize when block sparse attention can reliably recover the dominant attention mass. The key intuition is that tokens within the same block tend to share underlying semantic components, which induce coherent block-level structure.

**Assumption 4.1** (TOKEN REPRESENTATION MODEL). Let $\mathcal{B}_u$ and $\mathcal{B}_v$ denote the $u$-th query block and $v$-th key block, respectively. Each query and key vector is decomposed as

$$q_i = \bar{q}_u + \xi_u^{(q)} + \varepsilon_i^{(q)}, \qquad k_j = \bar{k}_v + \xi_v^{(k)} + \varepsilon_j^{(k)}, \quad (4)$$

where $i \in \mathcal{B}_u$ and $j \in \mathcal{B}_v$. Here, $\bar{q}_u$ and $\bar{k}_v$ are deterministic block centroids, $\xi^{(\cdot)}$ denote block-level semantic drift shared across tokens within a block, and $\varepsilon^{(\cdot)}$ capture token-level perturbations. We assume $\xi^{(\cdot)} \sim \mathcal{N}(0, \tau^2 I)$ and $\varepsilon^{(\cdot)} \sim \mathcal{N}(0, \sigma^2 I)$, independent across tokens and blocks.

Under this model, we analyze the block sparse attention paradigm that relies on block-wise pooled representations.

**Definition 4.2** (POOLED BLOCK CENTROIDS). For block $\mathcal{B}_u$ and $\mathcal{B}_v$ of size $B$, we define the pooled query and key centroids as

$$\hat{q}_u := \frac{1}{B} \sum_{i \in \mathcal{B}_u} q_i, \quad \hat{k}_v := \frac{1}{B} \sum_{j \in \mathcal{B}_v} k_j \quad (5)$$

For a fixed query block, the softmax normalization term is shared across all key blocks. Therefore, block selection can be approximated by ranking dot products between centroids.

**Definition 4.3** (APPROXIMATE BLOCK SCORE). We define the block-level score used for Top-$K$ selection as

$$\hat{S}_{uv} := \langle \hat{q}_u, \hat{k}_v \rangle, \tag{6}$$

and denote the selected blocks as

$$\hat{\mathcal{T}}_K := \arg \max_{v \in \{1,...,M\}}^{K} \hat{S}_{uv}. \tag{7}$$

To evaluate the accuracy of block selection, we compare $\hat{\mathcal{T}}_K$ with the oracle Top-$K$ blocks defined by exact attention mass. Specifically, Let

$$\mathcal{T}_K := \arg \max_{v \in \{1,...,M\}}^{K} \alpha_{uv}, \quad \alpha_{uv} := \sum_{i \in \mathcal{B}_u} \sum_{j \in \mathcal{B}_v} A_{ij}, \tag{8}$$

where $A_{ij}$ denotes the full attention matrix.

**Theorem 4.4** (PROBABILITY OF CORRECT BLOCK SELECTION). *Let*

$$\Delta \mu_{\min} := \min_{v \in \mathcal{T}_K, v' \notin \mathcal{T}_K} \langle \bar{q}_u, \bar{k}_v - \bar{k}_{v'} \rangle$$

*denote the minimum similarity gap between relevant and irrelevant block centroids, and assume $\|\bar{q}_u\|^2, \|\bar{k}_v\|^2 \leq C$. Then, there exists a constant c such that*

$$\mathbb{P}(\mathcal{T}_K = \hat{\mathcal{T}}_K) \geq 1 - \sum_{v \in \mathcal{T}_K} \sum_{v' \notin \mathcal{T}_K} \left[ e^{-\phi_1} + e^{-\phi_2} \right] \tag{9}$$

*where*

$$\phi_1 = \frac{\Delta \mu_{\min}^2}{48 C \delta^2}, \quad \phi_2 = c \min \left( \frac{\Delta \mu_{\min}^2}{\delta^4 d}, \frac{\Delta \mu_{\min}}{\delta^2} \right)$$

*and $\delta^2 = \tau^2 + \sigma^2/B$.*

The bound highlights that correct block recovery is primarily determined by the minimum centroid similarity gap. Larger token-level noise $\sigma^2$ and block-level semantic drift $\tau^2$ degrade selection reliability. Finally, we relate block selection accuracy to attention recall.

**Corollary 4.5** (EXPECTED ATTENTION RECALL). *Let $\gamma = K/M$ denote the sparsity budget. The expected attention recall for query block $\mathcal{B}_u$ satisfies*

$$\mathbb{E}[\mathcal{R}_u] \geq \gamma \cdot \mathbb{P}(\hat{\mathcal{T}}_K = \mathcal{T}_K). \tag{10}$$

### 4.3. Determinants of effective block sparse attention

From Corollary 4.5, the key factors affecting the performance of block sparse attention are the sparsity budget, the minimum inter-block similarity gap, the token-level perturbations, and the block-level drift. The derived lower bound of attention recall motivates us to propose an optimized block sparse attention through the following aspects: enlarging the inter-block similarity gap, reducing the block semantic diversity, and reallocating the sparsity budget.

---

**Algorithm 1** DFSAttn

**Input:** $Q, K, V \in \mathbb{R}^{N \times d}$, block size $B$, sub-block size $B_s$, permutation $\mathcal{P}$, sparsity budget $\gamma_t$, cached mask $\mathcal{M}$, mask update interval $\Delta$, diffusion step $t$

**Step 1: 3D Hilbert Curve Token Reordering**
$Q' \leftarrow \mathcal{P}(Q), K' \leftarrow \mathcal{P}(K), V' \leftarrow \mathcal{P}(V)$

**Step 2: Hierarchical Block Scoring**
**if** $t \equiv 0 \pmod{\Delta}$ **then**
  $\hat{Q} \leftarrow \text{AvgPool}(Q', B_s), \ \hat{K} \leftarrow \text{AvgPool}(K', B_s)$
  $\hat{A} \leftarrow \text{softmax}(\hat{Q}\hat{K}^T/\sqrt{d})$
  $\hat{S}_{uv} \leftarrow \text{Aggregation}(\hat{A}, B, B_s)$   ▷ Equation (11)
  $\mathcal{M}_t \leftarrow \text{TopK-Selection}(\hat{S}, \gamma_t)$   ▷ Equation (12)
  $\mathcal{M} \leftarrow \mathcal{M}_t$   ▷ Update cached mask
**else**
  $\mathcal{M}_t \leftarrow \mathcal{M}$   ▷ Reuse cached mask
**end if**

**Step 3: Block Sparse Attention**
$O' \leftarrow \text{SparseFlashAttn}(Q', K', V', \mathcal{M}_t)$
$O \leftarrow \mathcal{P}^{-1}(O')$   ▷ Restore original order
**Output:** $O$

---

## 5. Method

In this section, we introduce DFSAttn, a training-free dynamic sparse attention method for accelerating video generation in diffusion transformers. As illustrated in Figure 1, DFSAttn comprises three key strategies that jointly improve efficiency without compromising generation quality. First, we reorder video tokens using a 3D Hilbert curve, implicitly inducing fine-grained sparsity while preserving efficient GPU execution(Section 5.1). Second, we introduce hierarchical block score estimation, which refines block-level importance approximation via fine-grained sub-block aggregation (Section 5.2). Third, we cache and reuse sparse masks across diffusion steps with an adaptive sparsity budget, further reducing the computational overhead (Section 5.3).

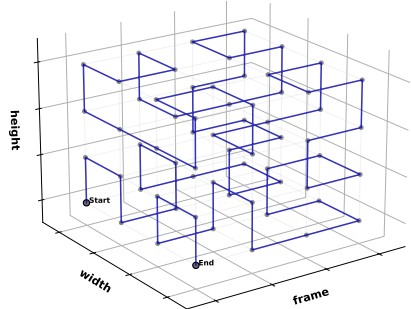

*Figure 4.* The 3D Hilbert curve in $4 \times 4 \times 4$ space.

*Table 1.* Reordering reduces the intra-block variance of queries and keys.

|  | Var($q$) | Var($k$) |
|---|---|---|
| w/o reordering | 1.22 | 1.25 |
| w reordering | 0.98 | 1.02 |

### 5.1. 3D Hilbert curve token reordering

As analyzed in Section 4.3, enlarging the inter-block similarity gap is critical for effective block sparse attention. Equivalently, tokens assigned to the same block should be as semantically coherent as possible. In video data, semantic relevance is strongly correlated with spatial–temporal proximity: tokens corresponding to nearby pixels across adjacent frames tend to share similar semantics. However, DiTs flatten 3D video tokens to a 1D sequence using per-frame row-major ordering, which disrupts the locality. Consequently, tokens that are spatially or temporally adjacent in the original video may be far apart in the sequence, leading to scattered attention patterns, as observed in Figure 2.

To address this mismatch, we reorder video tokens using a 3D Hilbert space-filling curve (Hilbert, 1891). The Hilbert curve is a continuous fractal curve with strong locality-preserving properties and has been widely adopted in vision models (Li & Xu, 2025; Zheng et al., 2025). While Jenga (Zhang et al., 2026) introduces Hilbert reordering applied recursively within each block, our approach performs a global reordering directly over the entire video tensor without any block-level decomposition. As illustrated in Figure 4, this mapping preserves spatial–temporal neighborhoods when projecting 3D video tokens to a 1D sequence.

Formally, as detailed in Algorithm 1, we define the reordering as a token permutation $\mathcal{P}$. Tokens are reordered at the beginning of each transformer block, block sparse attention is applied to the reordered sequence, and the inverse permutation $\mathcal{P}^{-1}$ is applied to the attention output. This design ensures that spatial–temporal neighboring tokens-typically those with high semantic relevance—remain adjacent in the 1D sequence. Consequently, each block in the reordered sequence corresponds to a coherent video region, while different blocks tend to capture distinct regions, naturally enlarging the inter-block similarity gap. Empirically, we further quantify the effect of reordering by measuring the average intra-block variance of queries and keys before and after applying 3D Hilbert reordering. As reported in Table 1, the proposed reordering strategy reduces intra-block variance by approximately 20% for both queries and keys, which is consistent with the intuition.

Critically, this reordering induces fine-grained sparsity while retaining GPU-friendly block-sparse operations. Although prior work such as SpargeAttn (Zhang et al., 2025a)

employs the Hilbert curve for sparse mask identification, it does not alter the contiguous block partitioning of the original sequence. In contrast, as illustrated in Figure 1, block-level sparsity applied to the reordered sequence translates into a substantially finer-grained sparsity pattern in the original spatiotemporal space.

### 5.2. Hierarchical block scoring

As shown in Section 4.3, reducing block-level semantic drift improves the lower bound on attention recall, highlighting the importance of accurate block-level representations in block sparse attention. However, existing methods typically construct block representations via coarse pooling, which implicitly assumes semantic homogeneity within each block. When a block contains multiple semantic centroids, pooling forces all tokens to be represented by a single vector, leading to inaccurate estimation of block-level importance.

To address this limitation, we propose a hierarchical block scoring strategy that aggregates fine-grained importance from sub-blocks. As detailed in Algorithm 1, each block is first partitioned into smaller sub-blocks, within which semantic variation is more locally coherent. We then compute sub-block-wise attention scores and aggregate them to form the block-level score. Formally, for a given attention head, let $\hat{A}$ denote the sub-block-wise attention score, $\{i', j'\}$ index sub-blocks within the query and key blocks. The block-level importance score is defined as

$$\hat{S}_{uv} = \sum_{i' \in \mathcal{B}_u} \sum_{j' \in \mathcal{B}_v} \hat{A}_{i' j'} \tag{11}$$

By performing aggregation over smaller sub-blocks, hierarchical block scoring yields a more faithful approximation of token-level attention interactions, leading to more reliable block ranking. For each query block, we rank key blocks by $\hat{S}_{uv}$ and select the most critical ones. Specifically, the selected block set $\mathcal{I}_u$ and the corresponding sparse mask are given by

$$\mathcal{I}_u = \arg \max_{v \in \{1,...,M\}}^{\gamma M} \hat{S}_{uv}, \quad \mathcal{M}[u, \mathcal{I}_u] = 1. \tag{12}$$

Finally, DFSAttn applies Flash Attention only to the block pairs specified by the sparse mask, significantly reducing the computational overhead of attention.

### 5.3. Sparse mask caching with adaptive ratio

As illustrated in Figure 3, under a fixed sparsity ratio, block sparse attention becomes increasingly effective as the diffusion process advances. Equivalently, achieving comparable accuracy at earlier diffusion steps requires a larger sparsity budget than at later steps. Leveraging this insight, we reallocate the overall computational budget across diffusion

*Table 2.* Quality and efficiency results of DFSAttn and other baselines.

| Method | Quality | | | | | | | | Efficiency | | |
|--------|---------|---|---|---|---|---|---|---|-----------|---|---|
| | PSNR ↑ | SSIM ↑ | LPIPS ↓ | AQ ↑ | BC ↑ | IQ ↑ | MS ↑ | SC ↑ | Sparsity ↑ | Latency ↓ | Speedup ↑ |
| **Wan2.1** | – | – | – | 59.77 | 95.09 | 65.49 | 98.72 | 94.35 | 0% | 1884s | 1.00× |
| Radial | 17.405 | 0.624 | 0.357 | 59.29 | 95.23 | 64.27 | 98.79 | 94.01 | 73.78% | 1098s | 1.72× |
| SVG | 17.393 | 0.612 | 0.362 | 57.92 | 95.42 | 65.24 | 98.85 | 93.79 | 65.71% | 1079s | 1.75× |
| SVG2 | 18.034 | 0.640 | 0.338 | 57.71 | 95.20 | 63.28 | 98.78 | 93.75 | 68.19% | 992s | 1.90× |
| **Ours** | **22.370** | **0.764** | **0.183** | 58.61 | 94.86 | 64.97 | 98.72 | 93.91 | **78.51%** | 1050s | **1.79×** |
| **Hunyuan** | – | – | – | 57.75 | 96.75 | 65.95 | 99.34 | 96.87 | 0% | 1752s | 1.00× |
| Radial | 20.897 | 0.750 | 0.285 | 57.21 | 97.01 | 67.16 | 99.39 | 96.72 | 72.67% | 1006s | 1.74× |
| SVG | 26.825 | 0.853 | 0.141 | 56.86 | 97.10 | 65.89 | 99.41 | 96.64 | 75.24% | 911s | 1.92× |
| SVG2 | 28.577 | 0.864 | 0.139 | 55.10 | 96.77 | 63.13 | 99.36 | 96.42 | 71.88% | 796s | 2.20× |
| **Ours** | **29.381** | **0.898** | **0.087** | 57.18 | 96.47 | 66.06 | 99.32 | 96.73 | **80.53%** | 836s | **2.10×** |

steps by adopting an adaptive sparsity schedule, where the sparsity budget decreases monotonically over time.

To further improve the accuracy-efficiency trade-off, we introduce a sparse mask caching strategy inspired by prior work on caching intermediate features in DiTs (Zhao et al., 2025; Liu et al., 2025). As detailed in Algorithm 1, DFSAttn identifies block sparse attention masks for each layer and head in the initial iteration and subsequently updates them at a fixed interval. Importantly, while the sparse masks are cached and reused across diffusion steps, the corresponding sparse attention outputs are recomputed at every step, ensuring that the dynamic evolution of token representations is fully preserved.

By jointly applying budget reallocation and sparse mask caching, DFSAttn achieves higher generation quality while substantially accelerating inference, demonstrating a favorable balance between accuracy and efficiency.

# 6. Experiment

## 6.1. Setup

**Models and datasets.** We evaluate DFSAttn on two state-of-the-art text-to-video diffusion models: HunyuanVideo-T2V-13B (Kong et al., 2024) and Wan2.1-T2V-14B (Wan et al., 2025), generating 129-frame and 81-frame videos at 720p resolution, respectively. All experiments use text prompts from the Penguin Benchmark (Kong et al., 2024).

**Metrics.** We measure the fidelity of sparse attention outputs relative to full attention using PSNR, SSIM, and LPIPS. In addition, we assess the video quality with VBench (Huang et al., 2024), reporting aesthetic quality (AQ), background consistency (BC), imaging quality (IQ), motion smoothness (MS) and subject consistency (SC). The efficiency of sparse attention methods is quantified by sparsity, defined as the fraction of attention computations eliminated compared to full attention.

**Baselines.** We compare DFSAttn with state-of-the-art sparse attention methods, including static approaches SVG (Xi et al., 2025) and RadialAttention (Li et al., 2026), as well as the dynamic method SVG2 (Yang et al., 2026).

**Implementation details.** We employ the Block-Sparse Attention kernel from (Guo et al., 2024) and benchmark DFSAttn on an NVIDIA H100 GPU with CUDA 12.4. The default attention backend is FlashAttention-2 (Dao, 2024). Following prior work (Li et al., 2026; Xi et al., 2025), sparse attention is bypassed during the first 25% of the denoising steps for all methods. Our adaptive sparsity budget is initialized at 0.3 and decreased by 0.1 every subsequent 25% of denoising steps, resulting in an average sparsity level of approximately 80% over the remaining steps. We use a block size of 128 and a sub-block size of 16. Since RadialAttention (Li et al., 2026) does not natively support 720p resolution, we pad video tokens to apply its attention mask. All baseline methods are evaluated using their official codebase, as detailed in Appendix B.

## 6.2. Quality and efficiency results

As shown in Table 2, DFSAttn consistently outperforms all baseline methods across similarity metrics: PSNR, SSIM, and LPIPS, even at higher sparsity levels. Notably, at approximately 80% sparsity, DFSAttn achieves an average PSNR of 22.37 on Wan2.1 and 29.38 on HunyuanVideo, demonstrating its ability to preserve fidelity under extreme compression. Corresponding qualitative results in Figure 5 further illustrate that DFSAttn generates videos highly consistent with full attention, preserving fine-grained details and temporal consistency compared to baselines. On the VBench benchmark, DFSAttn closely matches full attention across all evaluation metrics, highlighting its ability to maintain overall video quality. Additional video samples are provided in Appendix D.

In terms of inference efficiency, DFSAttn delivers 2.1× and 1.8× end-to-end speedups on HunyuanVideo and Wan

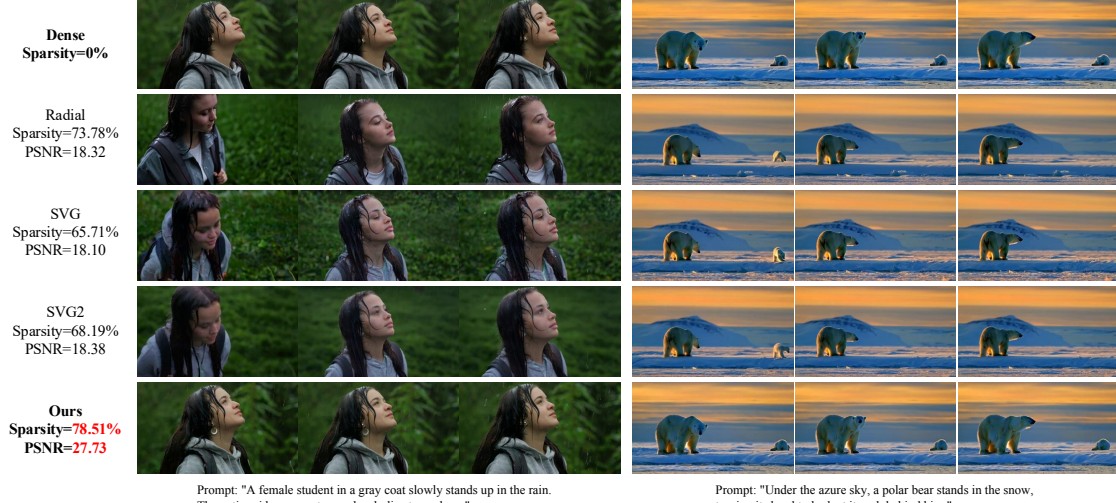

Dense
Sparsity=0%

Radial
Sparsity=73.78%
PSNR=18.32

SVG
Sparsity=65.71%
PSNR=18.10

SVG2
Sparsity=68.19%
PSNR=18.38

**Ours**
**Sparsity=78.51%**
**PSNR=27.73**

Prompt: "A female student in a gray coat slowly stands up in the rain. The entire video presents a melancholic atmosphere."

Prompt: "Under the azure sky, a polar bear stands in the snow, turning its head to look at its cub behind him."

*Figure 5.* Examples of videos generated by DFSAttn and other baselines on Wan2.1-T2V-14B.

*Table 3.* Ablation of token reordering on HunyuanVideo.

| Method | PSNR ↑ | SSIM ↑ | LPIPS ↓ |
|---|---|---|---|
| Raster | 27.794 | 0.874 | 0.124 |
| Hilbert2D | 29.265 | 0.893 | 0.090 |
| Block3D | 29.156 | 0.897 | 0.090 |
| Hilbert3D | **29.378** | **0.901** | **0.087** |

*Table 4.* Ablation of sub-block size on HunyuanVideo.

| Sub-block size | PSNR↑ | SSIM↑ | LPIPS↓ | Latency |
|---|---|---|---|---|
| 16 | **29.378** | **0.901** | **0.087** | 830.4s |
| 32 | 29.151 | 0.898 | 0.089 | 828.9s |
| 64 | 28.894 | 0.892 | 0.092 | 827.8s |
| w/o sub-block | 28.565 | 0.887 | 0.098 | 827.3s |

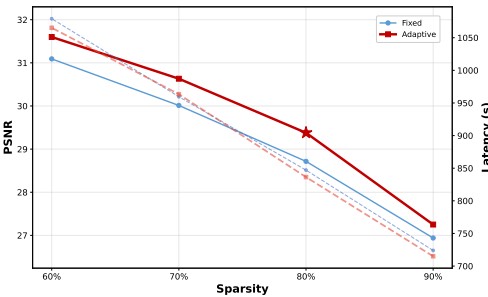

*Figure 6.* PSNR (solid, left) and latency (dashed, right) vs. overall sparsity. Main results are evaluated under adaptive 80% sparsity.

2.1, respectively, outperforming RadialAttention and SVG. Although SVG2 achieves slightly higher speedups with customized kernels, it suffers from a substantial degradation in visual quality. For example, DFSAttn achieves imaging quality scores of 66.06 and 64.97 on HunyuanVideo and Wan2.1, whereas SVG2 attains only 63.13 and 63.28, respectively. Additionally, we evaluate the scalability of DFSAttn under resolutions and frames in Appendix C.1. These results demonstrate that DFSAttn consistently maintains high fidelity under higher sparsity while delivering substantial acceleration, effectively balancing efficiency and quality.

### 6.3. Ablation study

**Token reordering.** Beyond the default Raster scan, we evaluate two alternative strategies: *Hilbert2D*, which applies Hilbert ordering independently within each frame, and *Block3D*, which partitions tokens into $4 \times 4 \times 4$ cubes

followed by local raster traversal. As shown in Table 3, locality-aware reordering consistently outperforms the standard Raster scan across all metrics. Among the evaluated strategies, Hilbert3D achieves the best performance by jointly preserving spatial and temporal locality throughout the sequence. In contrast, Hilbert2D ignores temporal continuity across frames, while Block3D introduces artificial partition boundaries that hinder global locality preservation. These results indicate that effective token traversal orders should maintain spatio-temporal locality simultaneously, which is precisely the design principle of Hilbert3D. The computational overhead of reordering is minimal in practice. The traversal order is computed once per inference and implemented as a static index mapping. For sequences containing approximately 120K tokens, the reordering overhead accounts for only 2% of total runtime, which is negligible relative to the efficiency gains.

**Sub-block size.** Table 4 evaluates the effect of sub-block granularity of hierarchical block scoring. Smaller sub-

blocks consistently improve reconstruction quality, as finer partitioning provides more accurate block-level representations for estimating attention importance. Notably, latency remains largely unchanged across all settings, suggesting that finer sub-block partitioning introduces negligible additional overhead. Additionally, we fix the block size to 128 throughout all experiments to align with GPU kernel design and maximize sparse attention execution efficiency.

**Mask update interval.** In the main experiments, we update the sparse attention mask every 25% of the total denoising steps. This setting preserves comparable video quality while reducing inference latency compared with per-step mask recomputation. The update interval is not fixed, and can be adjusted according to the sampling schedule. For example, in reduced-step distilled models, the shorter sampling trajectory naturally limits the opportunities for mask reuse; nevertheless, DFSAttn still provides per-step acceleration through sparse attention computation. We provide additional results on a 4-step distilled model in Appendix C.2.

**Adaptive sparsity budget.** Figure 6 presents the trade-off between video quality and inference latency under different overall sparsity budgets. We compare a fixed sparsity schedule, which applies a constant sparsity ratio throughout denoising, with the proposed adaptive schedule, where sparsity is dynamically adjusted across diffusion steps. As sparsity increases from 60% to 90%, both methods exhibit the expected degradation in PSNR together with reduced latency. However, the adaptive strategy consistently achieves superior reconstruction quality at comparable runtime, indicating that reallocating the sparsity budget across denoising steps substantially improves the effectiveness of block sparse attention. Based on this trade-off, we adopt the adaptive 80% sparsity setting as the default configuration for the main experiments.

## 7. Conclusion

In this paper, we identify the fundamental challenges that limit the effectiveness of block sparse attention in high-sparsity regimes induced by the dynamic and fine-grained attention patterns of diffusion transformers. Guided by the derived theoretical lower bound of attention recall, we propose DFSAttn, a training-free sparse attention framework that integrates 3D Hilbert curve–based token reordering, hierarchical block scoring, and sparse mask caching with adaptive ratio, enabling fine-grained sparsification while preserving efficient GPU execution. Extensive experiments demonstrate that DFSAttn consistently outperforms existing sparse attention methods, achieving superior generation quality under a high sparsity level.

## Acknowledgements

This work is funded by the National Key Research and Development Program of China (No. 2024YFA1012902) and the National Natural Science Foundation of China (No. W2441021, 12288101). This research is also supported by the AI for Science Institute, Beijing, China and the National Engineering Laboratory for Big Data Analytics and Applications.

## Impact Statement

This paper presents work whose goal is to advance the field of Machine Learning. There are many potential societal consequences of our work, none which we feel must be specifically highlighted here.

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

## A. Proof of Theorem 4.4 and Corollary 4.5

Before proving Theorem 4.4, we need a few lemmas.

**Lemma A.1** (Pooled Centroid Distribution). *Under Assumption 4.1, The pooled block centroids satisfy*

$$\hat{q}_u = \bar{q}_u + \xi_u^{(q)} + \bar{\varepsilon}_u^{(q)}, \quad \hat{k}_v = \bar{k}_v + \xi_v^{(k)} + \bar{\varepsilon}_v^{(k)} \tag{13}$$

*where $\bar{\varepsilon}_u^{(q)} \sim \mathcal{N}\left(0, \frac{\sigma^2}{B}I\right), \bar{\varepsilon}_v^{(k)} \sim \mathcal{N}\left(0, \frac{\sigma^2}{B}I\right).$*

*Proof.* By definition,

$$\hat{q}_u = \frac{1}{B} \sum_{i \in \mathcal{B}_u} \left(\bar{q}_u + \xi_u^{(q)} + \varepsilon_i^{(q)}\right) = \bar{q}_u + \xi_u^{(q)} + \frac{1}{B} \sum_{i \in \mathcal{B}_u} \varepsilon_i^{(q)}. \tag{14}$$

Defining $\bar{\varepsilon}_u^{(q)} := \frac{1}{B} \sum_{i \in \mathcal{B}_u} \varepsilon_i^{(q)}$, by independence and $\varepsilon_i^{(q)} \sim \mathcal{N}(0, \sigma^2 I)$, we have $\bar{\varepsilon}_u^{(q)} \sim \mathcal{N}(0, \frac{\sigma^2}{B}I)$. The derivation for $\hat{k}_v$ is analogous. □

**Lemma A.2** (Expectation of Approximate Block Score). *Under Assumptions 4.1:*

$$\mathbb{E}[\hat{S}_{uv}] = \langle \bar{q}_u, \bar{k}_v \rangle \tag{15}$$

*Proof.* Let

$$\zeta_u^{(q)} := \xi_u^{(q)} + \bar{\varepsilon}_u^{(q)}, \quad \zeta_v^{(k)} := \xi_v^{(k)} + \bar{\varepsilon}_v^{(k)} \tag{16}$$

According to Assumption 4.1, $\zeta_u^{(q)}$ and $\zeta_v^{(k)}$ are independent and they satisfy:

$$\zeta_u^{(q)} \sim \mathcal{N}\left(0, \delta^2 I\right), \quad \zeta_v^{(k)} \sim \mathcal{N}\left(0, \delta^2 I\right). \tag{17}$$

where $\delta^2 = \tau^2 + \frac{\sigma^2}{B}$. From Lemma A.1, we can expand $\hat{S}_{uv}$ as

$$\hat{S}_{uv} = \langle \bar{q}_u, \bar{k}_v \rangle + \langle \bar{q}_u, \zeta_v^{(k)} \rangle + \langle \bar{k}_v, \zeta_u^{(q)} \rangle + \langle \zeta_u^{(q)}, \zeta_v^{(k)} \rangle \tag{18}$$

Since all random terms are zero-mean and independent, the last three terms vanish in expectation. Therefore,

$$\mathbb{E}[\hat{S}_{uv}] = \langle \bar{q}_u, \bar{k}_v \rangle. \tag{19}$$

□

**Lemma A.3** (Pairwise score ordering). *Define $D_{vv'} := \hat{S}_{uv} - \hat{S}_{uv'}$ and $\Delta\mu_{vv'} := \langle \bar{q}_u, \bar{k}_v - \bar{k}_{v'} \rangle$. There exists a universal constant $c > 0$ such that*

$$\mathbb{P}(D_{vv'} < 0) \le \exp\left(-\frac{\Delta\mu_{vv'}^2}{8\left(\|\bar{k}_v - \bar{k}_{v'}\|^2 + 2\|\bar{q}_u\|^2\right)\delta^2}\right) + \exp\left(-c \min\left(\frac{\Delta\mu_{vv'}^2}{\delta^4 d}, \frac{\Delta\mu_{vv'}}{\delta^2}\right)\right). \tag{20}$$

*Proof.* From Equation (18), we can write

$$D_{vv'} = \langle \bar{q}_u, \bar{k}_v - \bar{k}_{v'} \rangle + \langle \bar{q}_u, \zeta_v^{(k)} - \zeta_{v'}^{(k)} \rangle + \langle \bar{k}_v - \bar{k}_{v'}, \zeta_u^{(q)} \rangle + \langle \zeta_u^{(q)}, \zeta_v^{(k)} - \zeta_{v'}^{(k)} \rangle. \tag{21}$$

By Lemma A.2, $\mathbb{E}[D_{vv'}] = \Delta\mu_{vv'}$. Let $\mu_D := \mathbb{E}[D_{vv'}]$ and write

$$D_{vv'} - \mu_D = \underbrace{\langle \bar{q}_u, \zeta_v^{(k)} - \zeta_{v'}^{(k)} \rangle + \langle \bar{k}_v - \bar{k}_{v'}, \zeta_u^{(q)} \rangle}_{\text{linear term}} + \underbrace{\langle \zeta_u^{(q)}, \zeta_v^{(k)} - \zeta_{v'}^{(k)} \rangle}_{\text{quadratic term}}. \tag{22}$$

We collect the Gaussian variables into

$$z = \begin{pmatrix} \zeta_u^{(q)} \\ \zeta_v^{(k)} \\ \zeta_{v'}^{(k)} \end{pmatrix} \in \mathbb{R}^{3d}, \quad z \sim \mathcal{N}(0, \Sigma), \quad \Sigma = \text{diag}(\delta^2 I_d, \delta^2 I_d, \delta^2 I_d). \tag{23}$$

Then the linear term can be written as $\ell^\top z$ for a deterministic vector $\ell$, and the quadratic term as $z^\top A z$, where

$$A = \frac{1}{2} \begin{pmatrix} 0 & I & -I \\ I & 0 & 0 \\ -I & 0 & 0 \end{pmatrix}.$$

Hence,

$$D_{vv'} - \mu_D = \ell^\top z + \left( z^\top A z - \mathbb{E}[z^\top A z] \right). \tag{24}$$

Applying a union bound,

$$\mathbb{P}(D_{vv'} < 0) \leq \mathbb{P}\left( \ell^\top z \leq -\frac{\mu_D}{2} \right) + \mathbb{P}\left( z^\top A z - \mathbb{E}[z^\top A z] \leq -\frac{\mu_D}{2} \right). \tag{25}$$

The first term is Gaussian with variance $\|\Sigma^{1/2}\ell\|_2^2$, yielding

$$
\begin{aligned}
\mathbb{P}\left( \ell^\top z \leq -\frac{\mu_D}{2} \right) &= \Phi\left( -\frac{\mu_D}{2\|\Sigma^{1/2}\ell\|_2} \right) \\
&\leq \exp\left( -\frac{\mu_D^2}{8\left( \|\bar{k}_v - \bar{k}_{v'}\|^2 + 2\|\bar{q}_u\|^2 \right)\delta^2} \right).
\end{aligned}
\tag{26}
$$

For the quadratic term, the Hanson–Wright inequality (Rudelson & Vershynin, 2013) implies that there exists a constant $c'$ such that

$$\mathbb{P}\left( \left( z^\top A z - \mathbb{E}[z^\top A z] \right) \leq -\frac{\mu_D}{2} \right) \leq \exp\left( -c' \min\left( \frac{\mu_D^2}{4K^4\|A\|_F^2}, \frac{\mu_D}{2K^2\|A\|} \right) \right). \tag{27}$$

where $K = \max_i \|z_i\|_{\psi_2}$ is the subgaussian norm of $z$, $\|A\|_F^2$ is the Frobenius norm of the matrix, and $\|A\|$ is the operator norm of the matrix. Since $z$ is Gaussian, the subgaussian norm satisfies

$$\|z_i\|_{\psi_2} \leq C_{\psi_2}\delta$$

for all coordinates $z_i$, where $C_{\psi_2} > 0$ is a universal constant. Combing $\|A\|_F^2 = \mathcal{O}(d)$ and $\|A\| = \mathcal{O}(1)$, By absorbing all constants into $c$, we have

$$\mathbb{P}\left( z^\top A z - \mathbb{E}[z^\top A z] \leq -\frac{\mu_D}{2} \right) \leq \exp\left( -c \min\left( \frac{\mu_D^2}{\delta^4 d}, \frac{\mu_D}{\delta^2} \right) \right), \tag{28}$$

Combining Equation (25), (26) and (28) completes the proof. □

Now we are ready to prove Theorem 4.4. We first restate the theorem.

**Theorem A.4** (PROBABILITY OF CORRECT BLOCK SELECTION). *Let*

$$\Delta\mu_{\min} := \min_{v \in \mathcal{T}_K, v' \notin \mathcal{T}_K} \langle \bar{q}_u, \bar{k}_v - \bar{k}_{v'} \rangle$$

*denote the minimum similarity gap between relevant and irrelevant block centroids, and assume $\|\bar{q}_u\|^2, \|\bar{k}_v\|^2 \leq C$. Then, there exists a constant $c$ such that*

$$\mathbb{P}(\mathcal{T}_K = \hat{\mathcal{T}}_K) \geq 1 - \sum_{v \in \mathcal{T}_K} \sum_{v' \notin \mathcal{T}_K} \left[ e^{-\phi_1} + e^{-\phi_2} \right] \tag{29}$$

*where*

$$\phi_1 = \frac{\Delta\mu_{\min}^2}{48C\delta^2}, \quad \phi_2 = c\min\left( \frac{\Delta\mu_{\min}^2}{\delta^4 d}, \frac{\Delta\mu_{\min}}{\delta^2} \right)$$

*and $\delta^2 = \tau^2 + \sigma^2/B$.*

*Proof.* By the definition in Theorem A.3, $\mathcal{T}_K = \hat{\mathcal{T}}_K$ if and only if:

$$D_{vv'} > 0 \quad \forall v \in \mathcal{T}_K, v' \notin \mathcal{T}_K.$$

Therefore, the probability that selected blocks align with the true top blocks is

$$
\begin{aligned}
\mathbb{P}(\mathcal{T}_K = \hat{\mathcal{T}}_K) &= \mathbb{P}\left( \bigcap_{v \in \mathcal{T}_K, v' \notin \mathcal{T}_K} \{D_{vv'} > 0\} \right) \\
&\geq 1 - \sum_{v \in \mathcal{T}_K} \sum_{v' \notin \mathcal{T}_K} \mathbb{P}(D_{vv'} \leq 0).
\end{aligned}
\tag{30}
$$

From Lemma A.3,

$$
\mathbb{P}(D_{vv'} < 0) \leq \exp\left( -\frac{\Delta\mu_{vv'}^2}{8\left( \|\bar{k}_v - \bar{k}_{v'}\|^2 + 2\|\bar{q}_u\|^2 \right) \delta^2} \right) + \exp\left( -c\min\left( \frac{\Delta\mu_{vv'}^2}{\delta^4 d}, \frac{\Delta\mu_{vv'}}{\delta^2} \right) \right).
\tag{31}
$$

Since $\|\bar{q}_u\|^2, \|\bar{k}_v\|^2 \leq C$, based on the definition of $\Delta\mu_{\min}$:

$$
\mathbb{P}(D_{vv'} < 0) \leq \exp\left( -\frac{\Delta\mu_{\min}^2}{48C\delta^2} \right) + \exp\left( -c\min\left( \frac{\Delta\mu_{\min}^2}{\delta^4 d}, \frac{\Delta\mu_{\min}}{\delta^2} \right) \right).
\tag{32}
$$

Combining Equation (30) and (32) completes the proof. $\square$

**Corollary A.5** (EXPECTED ATTENTION RECALL). *Let $\gamma = K/M$ denote the sparsity budget. The expected attention recall for query block $\mathcal{B}_u$ satisfies*

$$\mathbb{E}[\mathcal{R}_u] \geq \gamma \cdot \mathbb{P}(\hat{\mathcal{T}}_K = \mathcal{T}_K).
\tag{33}$$

*Proof.* Since the attention weights is non-negative, we have

$$\mathbb{E}\left[\mathcal{R}_u\right] \geq \mathbb{E}\left[\mathcal{R}_u | \mathcal{T}_K = \hat{\mathcal{T}}_K\right] \mathbb{P}(\mathcal{T}_K = \hat{\mathcal{T}}_K)
\tag{34}$$

According to the definition of true Top-K blocks in Equation (8):

$$\mathbb{E}\left[\mathcal{R}_u | \mathcal{T}_K = \hat{\mathcal{T}}_K\right] = \frac{\sum_{v \in \mathcal{T}_K} \alpha_{uv}}{\sum_{v=1}^{M} \alpha_{uv}} \geq \frac{K}{M}
\tag{35}$$

Combine with Theorem 4.4 yields the conclusion. $\square$

## B. Implementation Details

All baseline methods are evaluated using their official codebases and recommended settings. For RadialAttention (Li et al., 2026), the decay factor is set to be 0.95 for HunyuanVideo and 0.2 for Wan2.1. For SVG (Xi et al., 2025), the sparsity parameter for constructing the attention mask is set to 0.3 on Wan2.1 and 0.25 on HunyuanVideo. For SVG2, we adopt the recommended clustering parameters $C_q = 100, C_k = 500$, with a top-$p$ sparsity parameter of 0.9 to control token selection. The reported sparsity levels of baselines are provided by their official codebases.

As for DFSAttn, in practice, we set the sparse mask update interval to be 12 (approximately 25% of the 50 diffusion steps in the main experiments), and skip the sparsification of the first transformer block on Wan2.1. For end-to-end speedup comparison, we integrates the fast QK-Norm and RoPE CUDA kernels from Sparse-VideoGen.[1]

---

[1] https://github.com/svg-project/Sparse-VideoGen

*Table 5.* Performance of DFSAttn across resolutions and frames on HunyuanVideo.

| height × width × frame | PSNR ↑ | SSIM ↑ | LPIPS ↓ | Speedup ↑ |
|---|---|---|---|---|
| 544 × 960 × 129 | 28.3176 | 0.9059 | 0.0784 | 1.61× |
| 720 × 1280 × 65 | 29.7371 | 0.9174 | 0.0786 | 1.57× |
| 720 × 1280 × 129 | 28.6809 | 0.8952 | 0.0804 | 1.81× |
| 720 × 1280 × 193 | 29.6944 | 0.8906 | 0.0862 | 1.95× |

*Table 6.* Performance of DFSAttn on the Wan2.1 4-step distilled model. DFSAttn uses the same hyperparameter configuration as in the main experiments, with mask reuse disabled in this distilled setting.

| Method | AQ ↑ | BC ↑ | IQ ↑ | MS ↑ | SC ↑ | Speedup ↑ |
|---|---|---|---|---|---|---|
| Full | 0.3750 | 0.9228 | 0.3618 | 0.9804 | 0.8540 | 1.00× |
| DFSAttn | 0.3742 | 0.9187 | 0.3580 | 0.9810 | 0.8523 | 1.19× |

# C. Additional Results

## C.1. Scalability

We evaluate DFSAttn under varying resolutions and frames, as shown in Table 5. The results demonstrate that DFSAttn maintains stable generation quality across all settings, while achieving an increasing speedup at larger scales. This trend is expected, as the computational overhead of attention scales quadratically with sequence length, making sparsification become more pronounced for higher resolutions and longer videos. These results demonstrate that DFSAttn scales effectively without compromising visual quality, highlighting its suitability for high-resolution and long-video generation. Notably, the experiments here are conducted on an NVIDIA A100 GPU, which delivers smaller overall speedups compared to the H100 GPU in the main experiments.

## C.2. Results on Distilled Models

Many production-oriented video diffusion models employ step distillation or consistency distillation to reduce the number of denoising steps at inference time. This setting differs from standard multi-step sampling, where DFSAttn can further benefit from reusing sparse attention masks across nearby denoising steps. In reduced-step models, the shortened sampling trajectory naturally decreases the opportunities for mask reuse. We therefore evaluate whether DFSAttn remains effective when applied to a distilled model where acceleration mainly comes from per-step sparse attention computation.

Specifically, we evaluate DFSAttn on the 4-step distilled Wan2.1 model released by LightX2V.[2] We use the same DFSAttn hyperparameters as in the main experiments, while disabling mask reuse in this distilled setting. Sparse attention is bypassed at the first denoising step, and the sparse mask is recomputed every three subsequent steps using the adaptive sparsity budget.

Table 6 reports the results. DFSAttn achieves a 1.19× speedup over full attention on the 4-step distilled model, while maintaining comparable generation quality across all evaluated metrics. The changes in AQ, BC, IQ, and SC are minor, and MS remains on par with full attention. These results show that DFSAttn is compatible with distilled video diffusion models. Although reduced-step sampling limits the potential benefit of mask reuse, the per-step acceleration from sparse attention remains effective and provides complementary inference speedup without retraining or modifying the distilled checkpoint.

# D. Examples of Generated Videos

We provide several examples of videos generated by DFSAttn and full attention on HunyuanVideo and Wan2.1 in Figure 7 and Figure 8 respectively. The visualization further demonstrate that DFSAttn generates videos highly consistent with full attention, preserving fine-grained details and temporal consistency.

---

[2] https://huggingface.co/lightx2v/Wan2.1-Distill-Models

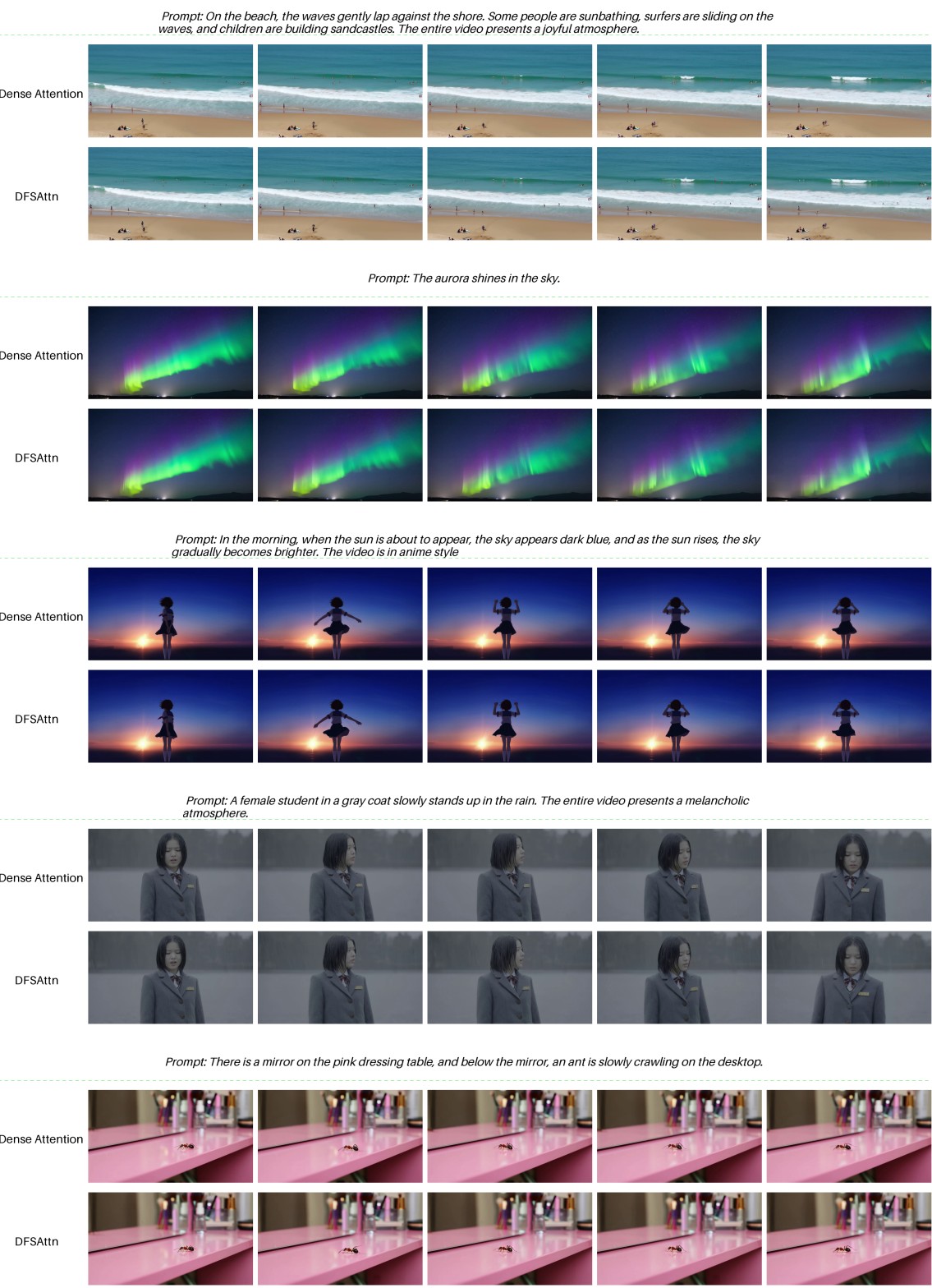

*Figure 7.* Video generation examples from dense attention and DFSAttn on HunyuanVideo.

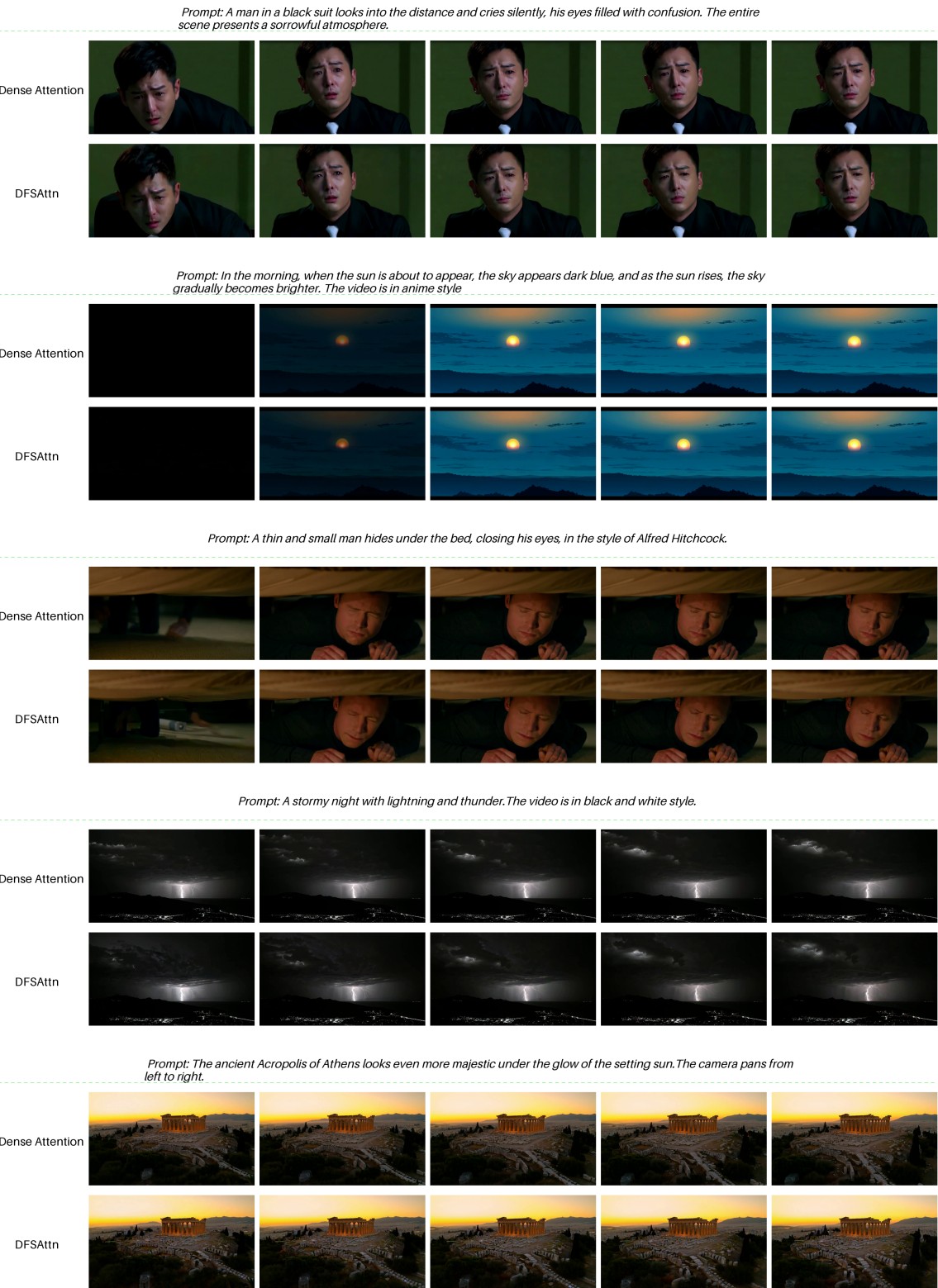

*Figure 8.* Video generation examples from dense attention and DFSAttn on Wan2.1-T2V-14B.

