# OpenReview forum: "DFSAttn: Dynamic Fine-grained Sparse Attention for Efficient Video Generation"
_ICML.cc/2026/Conference — ICML 2026 regular_

### Official Review · Reviewer_DSVV · 2026-03-04

**Soundness:** 3
**Presentation:** 3
**Significance:** 2
**Originality:** 3
**Overall Recommendation:** 4
**Confidence:** 4

**Summary:**

This paper proposes DFSAttn, a training-free sparse attention framework for accelerating video diffusion transformers. DFSAttn introduces three main components: (1) 3D Hilbert curve token reordering to preserve spatiotemporal locality and improve block coherence, (2) hierarchical block scoring that aggregates sub-block attention scores for more accurate block importance estimation, and (3) sparse mask caching with an adaptive sparsity schedule across diffusion timesteps. Experiments on HunyuanVideo and Wan2.1 show that DFSAttn maintains visual fidelity under high sparsity levels while achieving up to 2.1× end-to-end speedup. The method also improves reconstruction fidelity metrics such as PSNR, SSIM, and LPIPS compared to existing sparse attention baselines.

**Compliance With Llm Reviewing Policy:**

Affirmed.

**Final Justification:**

The paper is technically solid, with clear ablations and practical improvements, and overall it’s easy to follow . I had concerns about novelty, missing baselines, and how well it works with distilled models. The rebuttal does a good job adding baselines, but the novelty and real-world relevance are still not fully convincing to me. Overall, this feels like a useful but somewhat incremental paper, so I keep a weak accept.

**Key Questions For Authors:**

1. The paper reports experiments generating 129-frame videos for HunyuanVideo. However, the default configuration of HunyuanVideo uses 121 frames. Why is that?

**Limitations:**

no explicit discussion on limitation in the paper.

**Strengths And Weaknesses:**

**Strengths**

1. Soundness: The paper includes clear ablation studies isolating the impact of the main design components. In particular, the ablations demonstrate the benefit of Hilbert curve token reordering and hierarchical block scoring, showing consistent improvements in reconstruction metrics.

2. Presentation: The paper provides a theoretical analysis relating attention recall, sparsity budget, and block semantic diversity. This analysis helps motivate the design choices and provides a useful perspective on why block sparse attention may fail in video diffusion models.

3. Significance/Originality: see weakness

**Weakness**

1. Originality: The use of Hilbert curve reordering and block sparse attention is not entirely new and has been explored in prior work such as SpargeAttention. I would say only the hierarchical block scoring is something new.

2. Soundness: The evaluation does not include comparisons against several closely related sparse attention approaches. In particular, methods such as SpargeAttention and Video Sparse Attention would be important baselines for Table 1. Without these comparisons, it is difficult to fully assess the improvement over prior work.

3. Significance: The proposed method focuses on training-free sparse attention. However, many production-ready video diffusion models rely heavily on distillation techniques (e.g., step distillation or consistency distillation) to reduce sampling steps. The paper does not evaluate whether DFSAttn remains effective or compatible with such distilled models, escpetially for the mask caching part. This limits the practical relevance of the approach in 2026.

---

> ### Author Rebuttal · Authors · 2026-03-30
>
> We sincerely appreciate Reviewer DSVV for the valuable comments. The concerns are addressed as follows.
>
> ---
> **W1. The use of Hilbert curve reordering and block sparse attention is not entirely new and has been explored in prior work. I would say only the hierarchical block scoring is something new.**
>
> We agree that individual components such as Hilbert curve reordering and block sparse attention have been explored in prior work. However, our contribution goes beyond a direct combination of existing techniques.
>
> Specifically, we identify a key limitation of prior methods: the mismatch between coarse block sparsity and the dynamic, fine-grained attention patterns in DiTs. Based on this insight, we derive a theoretical lower bound that directly motivates three tightly coupled designs: 3D Hilbert reordering, hierarchical block scoring, and adaptive sparsification with mask caching. Importantly, these components work together to address the same underlying issue, leading to significantly improved performance under high sparsity.
>
> ---
> **W2. Additional baselines such as SpargeAttention.**
>
> We thank the reviewer for the suggestion to include more baselines. We have incorporated SpargeAttention [1] into our evaluation, and the results are presented in **Table G**. DFSAttn outperforms SpargeAttn in both quality and efficiency, even under higher sparsity ratios, demonstrating the effectiveness of our method.
>
> **Table G.** Comparsion with SpargeAttention on HunyuanVideoT2V-13B (A100).
>
> | Method | PSNR $\uparrow$ | SSIM $\uparrow$ | LPIPS $\downarrow$ | Sparsity | Speedup |
> | - | - | - | - | - | - |
> | SpargeAttn | 22.1068 | 0.7678 | 0.3822 | 70% | 1.52x |
> | DFSAttn | **28.6809** | **0.8952** | **0.0804** | 80% | 1.81x |
>
> [1] SpargeAttention: Accurate and Training-free Sparse Attention Accelerating Any Model Inference
>
> ---
> **W3. Compatibility with distilled models.**
>
> We appreciate the reviewer’s emphasis on compatibility with distilled models. DFSAttn is orthogonal to distillation techniques, as it accelerates attention within each denoising step without modifying the sampling procedure. For mask caching, although distilled models use fewer steps, temporal consistency between adjacent steps still exists. Our caching strategy can adapt by adjusting the update frequency, allowing it to remain effective even with shorter trajectories. While the total gain may decrease due to fewer steps, the per-step acceleration benefit is preserved.
>
> We agree that empirical validation on distilled models would further strengthen this point, and we will include such experiments and discussion in the revision.
>
> ---
> **Q1.The paper reports experiments generating 129-frame videos for HunyuanVideo. However, the default configuration of HunyuanVideo uses 121 frames. Why is that?**
>
> Thank you for pointing out this discrepancy. Our experiments follow the official implementation settings provided in Tencent-Hunyuan/HunyuanVideo repository, where the default configuration uses 129 frames.

---

> > ### Author Rebuttal · Reviewer_DSVV · 2026-04-01
> >
> > My concerns on W2 and Q1 is fully addressed and remain reserved for W1 and W3. I would thus keep my rating.

---

### Official Review · Reviewer_PFc5 · 2026-03-09

**Soundness:** 2
**Presentation:** 3
**Significance:** 2
**Originality:** 2
**Overall Recommendation:** 4
**Confidence:** 3

**Summary:**

This paper aims to enhance the inference efficiency of the diffusion transformer in generating high-quality videos. The authors pointed out that the quadratic complexity of the spatio-temporal three-dimensional full attention would lead to high computational costs, and they focused on the inherent sparsity patterns in the attention map of DiTs. However, the existing sparse attention methods rely on coarse-grained or static masks, which do not match the dynamic and fine-grained sparsity patterns of DiTs, and significantly degrade the quality at high sparsity ratios. The authors propose the DFSAttn method that does not require training to solve the balance problem between generation quality and computational efficiency. The proposed DFSAttn method combines token reordering based on a 3D Hilbert curve to achieve hardware-friendly fine sparsification, hierarchical block scoring, importance assessment, and adaptive sparse mask caching. The experiments in the paper show that the proposed DFSAttn method can accelerate the generation process and maintain a high level of visual fidelity.

**Compliance With Llm Reviewing Policy:**

Affirmed.

**Final Justification:**

The authors have addressed most of my concerns, and I will increase the rating.

**Key Questions For Authors:**

Here are my key questions.

(1) The authors should discuss how the sparse mask based on visual features constructed by DFSAttn interacts with the text condition, which interests me quite a bit. Do the unconditional generation branch and the conditional generation branch share the same sparse mask? I think that if they do, there might be potential risks, such as damaging the text alignment ability. If the authors could clarify the calculation strategy of the mask or provide relevant ablation experiments to prove that it does no harm to text consistency, it would alleviate my concerns.

(2) As discussed in the paper, the dense full attention mode transitions to a highly sparse state. Here, I believe it will bring instability, such as causing changes in the distribution of latent features. Does the author have any further explanation for this transition? If the authors can provide ablation experiments on the proportion of initial skipping steps or explain why a smooth transition mechanism is not necessary, it will enhance my confidence in the robustness of this method.

(3) In complex video dynamic physical scenes, is the three-dimensional Hilbert curve based on the static spatio-temporal proximity assumption still applicable for reordering? Could the authors explain if there are any failure cases in high-motion scenes and how to address them? If the authors could supplement generation cases in high-motion scenes and discuss the boundaries of this method, it would help me evaluate it more comprehensively.

(4) The authors claim that the proposed method holds good stability at high sparsity. However, in highly dynamic videos, such as when the camera moves rapidly, I think significant changes would occur within 12 diffusion steps, which could have many impacts, such as potentially losing the features of newly emerged or rapidly moving objects. For such special cases, what solutions do the authors think should be adopted?

**Limitations:**

As discussed in the weakness part. I suggest the authors thoroughly revise the paper in the following aspects.

(1) Although the paper suggests that reordering the data according to the 3D Hilbert curve can make the computation more efficient and claims that this method is hardware-friendly, it lacks a detailed analysis of the memory usage and memory bandwidth bottlenecks. I believe that this approach will lead to a significant increase in memory bandwidth and latency when dealing with massive tokens, so the analysis of relevant data is very important.

(2) The authors proposed an adaptive proportion sparse mask caching mechanism. Assuming that the attention masks between adjacent denoising time steps have high similarity, they believe that setting an update every 12 steps will perform well in most smooth videos. However, I think that the compatibility with few-step samplers, if the model uses acceleration algorithms to generate an image, setting12 steps for an update means that the mask is hardly updated during the generation process, which may lead to a significant drop in quality.

(3) Although the authors proposed using VBench to evaluate the overall quality and compared it with SVG and RadialAttention, I believe that these high-level semantic metrics may not capture the underlying pixel-level flickering problems caused by sparse attention. Additionally, the paper lacks the authoritative overall distribution metric FVD in the field of video generation.

(4) In Table 3, the authors only conducted a simple ablation experiment on the sub-block size. I believe that the setting of block size not only has a significant impact on the overall method performance, but also holds a mutual influence between sub-block size, especially in terms of setting values. However, the authors did not explore the joint ablation experiment of the two terms.

**Strengths And Weaknesses:**

**Strengths**
Overall, this paper addresses the issue of efficient video generation. The overall approach of the paper is simple and clear, yet it has practical and efficient significance. The authors found that the existing sparse attention methods rely on static or coarse-grained block masks, while ignoring the intrinsic dynamic and fine attention patterns that are crucial for generating high-quality dynamic images. Therefore, the authors proposed a framework that does not require training. It captures fine sparsity by using three-dimensional Hilbert curve marking reordering and significantly improves the generation fidelity and inference efficiency at high sparsity rates by using a hierarchical block scoring with adaptive mask caching.

**Weaknesses**
I'm not an expert in this field, so please correct me if any parts of my understanding are wrong. There are a few weaknesses that stop me from giving a higher rating. However, I'm still willing to read the author's responses to decide the final ratings. My major concerns are as follows.

(1) The authors claim that DFSAttn is hardware-friendly. However, I think this friendliness seems to be closely related to certain specific NVIDIA architectures. The paper did not explore this friendliness on other, more diverse edge or consumer-grade hardware platforms, especially hardware without sparse characteristics.

(2) The authors used the three-dimensional Hilbert curve to reorder the tokens for achieving hardware-friendly fine-grained processing. However, I believe the computational cost involved in the dynamically generated three-dimensional Hilbert curve sorting is not clear. In actual computational costs, especially for the processing of large tensors, the situation may be very different from that assumed in the paper.

(3) The authors proposed the adaptive sparse mask cache technology, which can reuse the mask in multiple denoising steps to improve efficiency. But why it can be applied to highly dynamic video generation seems somewhat difficult to understand. I think it may not be applicable in scenarios where objects are moving rapidly, or the camera is moving rapidly, for example, it may cause spatial errors and feature disappearance.

(4) The authors pointed out that the omission of the text condition token is acceptable. However, I think there are still some issues regarding the details. For example, it is unclear how DFSAttn handles cross-attention. Is it subjected to sparse processing or maintained in a dense state? Additionally, how does the mask cache operate, as mentioned in Figure 1?

---

> ### Author Rebuttal · Authors · 2026-03-30
>
> We sincerely appreciate Reviewer PFc5 for the insightful feedback. Due to space constraints, we address related concerns jointly below.
>
> ---
> **W1. Clarification of "hardware-friendly".**
>
> We thank the reviewer for pointing out this ambiguity. Our use of “hardware-friendly” specifically refers to GPU execution efficiency, not general hardware portability. In particular, DFSAttn adopts contiguous block sparsity, which aligns well with GPU memory access patterns and kernel execution. We will clarify it in the revision.
>
> ---
> **W2 & L1. Computational cost and memory behavior of Hilbert reordering.**
>
> We appreciate the reviewer for raising this concern. In practice, 3D Hilbert reordering is computed once per inference and implemented as a static index mapping. As shown in Table 2 (paper), for sequences with ~120k tokens, the reordering overhead accounts for only ~2% of total runtime, which is negligible compared to the savings from sparse attention.
>
> Regarding memory behavior, the reordering is implemented as a gather operation. While large tensors will introduce some bandwidth cost, the resulting permuted tensor is stored contiguously, ensuring that downstream attention computations maintain efficient memory access without additional latency. We agree that a more explicit memory analysis would strengthen the paper, and will include detailed profiling in the revision.
>
> ---
> **W3 & Q4. Sparse mask caching under dynamic conditions.**
>
> We would like to clarify that diffusion steps correspond to iterative denoising on the same latent video, rather than temporal progression across frames. As a result, attention patterns typically evolve smoothly across steps, even for visually dynamic content, enabling effective mask reuse.
>
> We acknowledge that in extreme cases, attention patterns may shift more significantly. In such scenarios, the mask update frequency can be increased, or adaptive refresh strategies (e.g., based on attention change) can be employed.
>
> ---
> **W4 & Q1. Interaction with text conditioning.**
>
> We apologize for the lack of clarity in the paper. We would like to clarify that:
> - Cross-attention remains dense in DFSAttn. Sparsification is applied only to visual self-attention. Therefore, text–video alignment is preserved.
> - The conditional and unconditional branches do not share sparse masks. Each branch independently computes its own mask based on its respective inputs.
>
> ---
> **Q2. Transition from dense to sparse attention.**
>
> Following prior works, we skip the initial 25% denoising steps and then directly apply sparse attention. We further compare it with a smooth transition scheme that gradually increases sparsity, ensuring the same overall budget.
>
> As shown in **Table F**, both strategies achieve comparable generation quality, while the direct transition yields slightly higher speedup due to reduced mask computation overhead. This suggests that a smooth transition is not necessary in practice.
>
> **Table F.** Ablation on transition strategy.
> |Method|PSNR$\uparrow$|SSIM$\uparrow$|LPIPS$\downarrow$|Speedup|
> |-|-|-|-|-|
> |smooth|28.6572|0.8956|0.0799|1.77x|
> |non-smooth|28.6809|0.8952|0.0804|1.81x|
>
> ---
> **Q3. Applicability in high-motion scenes.**
>
> We thank the reviewer for this insightful question. Indeed, the 3D Hilbert curve assumes that spatio-temporal proximity correlates with semantic similarity, which may not fully hold in high-motion scenarios.
>
> As shown in our supplementary results in https://anonymous.4open.science/r/case-5EAE, this may lead to artifacts such as missing parts or geometric distortions due to locality mismatch. Nevertheless, DFSAttn still maintains competitive performance overall. For such challenging scenarios, a more adaptive or motion-aware ordering strategy could be explored. We will include additional high-motion cases and discussion in the revision.
>
> ---
> **L2. Compatibility with few-step samplers.**
>
> We thank the reviewer for highlighting this important scenario. The mask update interval (e.g., every 12 steps) is not fixed and can be adjusted based on the number of denoising steps. For few-step samplers, we reduce the update interval accordingly to ensure that masks remain responsive to attention changes. While fewer steps naturally reduce the opportunity for reuse, the per-step acceleration benefit remains unchanged. We will include experiments and discussion for such settings.
>
> ---
> **L3. More evaluation metrics.**
>
> We thank the reviewer for this constructive suggestion. We will include additional metrics such as FVD to provide a more comprehensive assessment of visual quality in the revision.
>
> ---
> **L4. Joint ablation of block size and sub-block size.**
>
> We agree that block size and sub-block size jointly affect both efficiency and quality. In Table 3, we fix the block size based on efficiency consideration aligned with GPU kernel design. We will include joint ablation experiments and provide analysis on their interaction in the revision.

---

> > ### Author Rebuttal · Reviewer_PFc5 · 2026-04-02
> >
> > Dear authors, thank you for your responses. Most of my concerns have been addressed, and I will increase the score to 4.

---

### Official Review · Reviewer_2V5y · 2026-03-10

**Soundness:** 3
**Presentation:** 3
**Significance:** 3
**Originality:** 3
**Overall Recommendation:** 5
**Confidence:** 3

**Summary:**

This paper introduces a dynamic fine-grained sparse attention mechanism named DFSAttn, aiming to improve the inference efficiency of DiT in video generation. Addressing the issue of generation quality degradation in existing block sparse attention methods when applied to DiTs, the authors observe the dynamic nature of attention distribution during the diffusion process and accordingly derive a theoretical lower bound for attention recall. Guided by this theory, DFSAttn integrates Hilbert curve-based token reordering, a hierarchical block scoring mechanism, and a sparse mask caching strategy with adaptive ratios. This method accelerates end-to-end inference speed while maintaining high video generation quality.

**Compliance With Llm Reviewing Policy:**

Affirmed.

**Final Justification:**

I have read the authors' response and the other reviewers' comments, and I maintain my positive attitude.

**Key Questions For Authors:**

1. How does the Hilbert curve reordering actually perform in significantly reducing the theoretical intra-block variance?
2. Under extremely high sparsity ratios, will the model's multi-modal understanding capability for complex prompts be substantially impaired?
3. Does the actual acceleration ratio of the dynamically changing sparse mask caching strategy demonstrate general applicability across different hardware architectures?

**Limitations:**

The main limitations of this paper center on the simplified assumptions regarding complex data manifold distributions. In their assumptions, the authors simply model semantic variance and local perturbations as independent Gaussian distributions. Meanwhile, although highly concentrated sparse attention can accurately capture local features, it may inevitably cause the loss of some semantic information crucial for the reconstruction of fine-grained multi-modal features that depend on global structures.

**Strengths And Weaknesses:**

Strengths
1. The authors' observation of the inherent dynamics of the diffusion denoising process is highly insightful, providing valuable guidance for subsequent research.
2. This training-free architectural design possesses strong practical value.
3. By introducing Hilbert curve reordering and the hierarchical scoring mechanism, the authors effectively address the intra-block semantic variance and local perturbations identified in the theoretical formula from their physical roots. This makes the motivation for the practical design highly self-consistent.

Weaknesses

When the model's attention is forced into high-ratio sparsification, it may face the risk of degraded multi-modal semantic alignment capabilities when handling scenarios that require strong global context dependencies or complex fine-grained semantic interactions. The lack of specific evaluations on multi-modal understanding under high sparsity budgets makes the current method appear somewhat insufficiently comprehensive.

Furthermore, although the experimental section demonstrates end-to-end performance improvements, specific ablation studies targeting key variables in the theoretical lower bound—such as the actual change in intra-block variance after adopting the Hilbert curve—are insufficient. To some extent, this makes the chain of evidence between theory and experiment feel somewhat disjointed.

---

> ### Author Rebuttal · Authors · 2026-03-30
>
> We sincerely appreciate Reviewer 2V5y for the recognition of our work and the valuable comments. The concerns are addressed as follows. For weaknesses, please refer to the response to Q1 and Q2.
>
> ---
> **Q1. How does the Hilbert curve reordering perform in reducing the theoretical intra-block variance?**
>
> We appreciate the reviewer's insightful comment regarding a tighter connection between theory and empirical evidence. To this end, we directly measure the average intra-block variance of queries and keys before and after reordering.
>
> As shown in **Table C**, 3D Hilbert curve reordering reduces intra-block variance by approximately 20%. This empirically validates our theoretical analysis: by improving spatial-temporal locality, Hilbert reordering groups semantically similar tokens within each block, thereby tightening the variance-based lower bound and improving the fidelity of block sparse attention. This result strengthens the consistency between our theoretical analysis and practical design.
>
> **Table C.** Intra-block variance before and after reordering.
> | | Var(q) | Var(k) |
> | - | - | - |
> | w/o reordering | 1.22 | 1.25 |
> | w/ reordering | 0.98 | 1.02 |
>
> ---
> **Q2. Under extremely high sparsity ratios, will the model's multi-modal understanding capability for complex prompts be substantially impaired?**
>
> Thank you for raising this important concern. As expected, increasing sparsity reduces the capacity for modeling long-range and fine-grained interactions. As shown in Figure 6 (paper) and **Table D** below, very high sparsity levels (e.g., 90%) lead to noticeable performance degradation.
>
> However, we would like to emphasize that compared to prior sparse attention methods, DFSAttn consistently achieves better quality under the same sparsity budgets, indicating improved preservation of semantic information. It maintains strong performance at practical sparsity levels, achieving a favorable trade-off between efficiency and quality. In practice, the sparsity level can be adjusted based on task complexity, allowing DFSAttn to adapt to scenarios requiring stronger global modeling.
>
> **Table D.** Quality evaluation across sparsity ratios.
> | Sparsity | PSNR $\uparrow$ | SSIM $\uparrow$ | LPIPS $\downarrow$ |
> | - | - | - | - |
> | 70% | 30.6320 | 0.9178 | 0.0702 |
> | 80% | 29.3781 | 0.9012 | 0.0865 |
> | 90% | 27.2511 | 0.8587 | 0.1248 |
>
> ---
> **Q3. Does the acceleration generalize across hardware architectures?**
>
> Thank you for this practical question. As shown in **Table E**, DFSAttn achieves consistent end-to-end acceleration across different GPU architectures, including Ampere (A100) and Hopper (H100). While the achieved speedup varies due to architectural differences, our method consistently provides significant acceleration, demonstrating its general applicability.
>
> **Table E.** End-to-end speedups on different GPUs.
> | Model | A100 | H100 |
> | - | - | - |
> | Hunyuan | 1.81x | 2.10x |

---

> > ### Author Rebuttal · Reviewer_2V5y · 2026-04-03
> >
> > I thank the authors for their thorough response, which has fully addressed my concerns.

---

### Official Review · Reviewer_ZJVz · 2026-03-13

**Soundness:** 3
**Presentation:** 3
**Significance:** 3
**Originality:** 3
**Overall Recommendation:** 4
**Confidence:** 3

**Summary:**

The paper presents DFSAttn, which aims at improving video generation efficiency through Sparse Attention. The process includes (1) 3D Hilbert token reordering (2) Hierarchical block scoring (3) Sparse mask caching + adaptive sparsity schedule. As a result, the proposed DFSAttn achieves better accuracy-cost trade-off compared to prior methods, built upon HunyuanVideo and Wan2.1

**Compliance With Llm Reviewing Policy:**

Affirmed.

**Key Questions For Authors:**

- While the paper presents the re-ordering helps in tab2, I am wondering if the authors have tried other scanning orders? Any commons in the preferred orders?

- How is the method performance when scaling number of frames / resolutions?

- While the reported sparsity can reaches 80%, the real speed-up is merely 2.1x, what is the main bottleneck here?

**Limitations:**

yes

**Strengths And Weaknesses:**

- The paper targets at an important problem on improving the attention efficiency in video generation

- The proposed method is training-free and follows modular designs, making it easier to integrete into other methods / models.

- Evaluated across different models / benchmarks, the proposed method demonstrates reasonable good performance.

---

> ### Author Rebuttal · Authors · 2026-03-30
>
> We sincerely appreciate Reviewer ZJVz for the insightful questions. The concerns are addressed as follows.
>
> ---
> **Q1. Have the authors explored other scanning orders? Any commons in the preferred orders?**
>
> Thank you for raising this important point. In addition to the default Raster scanning used in DiTs and Hilbert3D adopted in our method, we evaluate two alternative orders: Hilbert2D (per-frame Hilbert ordering), and Block3D (partitioning tokens into 4×4×4 cubes with local raster ordering).
>
> As shown in **Table A**, reordering tokens to preserve locality consistently improves performance over Raster, enabling effective block sparse attention. Hilbert3D, adopted in our paper, outperforms the alternatives by jointly preserving spatial and temporal locality, whereas Hilbert2D neglects temporal adjacency and Block3D introduces artificial boundaries that limit global locality modeling. The results suggest that effective scanning orders should maximize locality preservation in both spatial and temporal dimensions, which aligns with the design of Hilbert3D.
>
> **Table A.** Ablation of token reordering.
> | Order | PSNR $\uparrow$ | SSIM $\uparrow$ | LPIPS $\downarrow$ |
> | - | - | - | - |
> | Raster | 27.7941 | 0.8739 | 0.1236 |
> | Hilbert2D | 29.2648 | 0.8927 | 0.0902 |
> | Block3D | 29.1555 | 0.8968 | 0.0896 |
> | Hilbert3D | **29.3781** | **0.9012** | **0.0865** |
>
> ---
> **Q2. How does the method perform when scaling number of frames / resolutions?**
>
> We evaluate DFSAttn under varying resolutions and frames, as shown in **Table B**. Experiments are conducted on an NVIDIA A100 GPU.
>
> The results demonstrate that DFSAttn maintains stable generation quality across all settings, while achieving increasing speedup at larger scales. This trend is expected, as the computational overhead of attention scales quadratically with sequence length, making sparsification become more pronounced for higher resolutions and longer videos. These results demonstrate that DFSAttn scales effectively without compromising visual quality, highlighting its suitability for high-resolution and long-video generation.
>
> **Table B.** Scalability across resolutions and frames.
> | Height x Width x Frame | PSNR $\uparrow$ | SSIM $\uparrow$ | LPIPS $\downarrow$ | Speedup |
> | - | - | - | - | - |
> | 544 x 960 x 129 | 28.3176 | 0.9059 | 0.0784 | 1.61x |
> | 720 x 1280 x 65 | 29.7371 | 0.9174 | 0.0786 | 1.57x |
> | 720 x 1280 x 129 | 28.6809 | 0.8952 | 0.0804 | 1.81x |
> | 720 x 1280 x 193 | 29.6944 | 0.8906 | 0.0862 | 1.95x |
>
> ---
> **Q3. What is the main bottleneck behind the gap between theoretical sparsity and the observed speedup?**
>
> The discrepancy arises from two primary factors:
> - The reported speedup measures the overall inference latency of the DiT pipeline, rather than the attention module alone. While DFSAttn substantially reduces attention cost, other components, such as linear layers and normalization, also contribute to total runtime.
> - Although sparsity reduces theoretical FLOPs, the actual speedup depends on GPU kernel efficiency. In current block-sparse implementations, factors such as memory access patterns and kernel scheduling limit full utilization of sparsity. Importantly, these limitations are common to existing sparse attention methods and not specific to DFSAttn.

---

> > ### Author Rebuttal · Reviewer_ZJVz · 2026-04-03
> >
> > I appreciate the rebuttal which fully addresses my concern, thus I will maintain my positive score.

---

### Decision · Program_Chairs · 2026-04-30

**Decision:**

Accept (regular)

**Comment:**

This paper introduces DFSAttn, a training-free sparse attention framework for accelerating video diffusion transformers. The reviewers generally agreed that it has clear technical and practical value. In particular, the paper’s analysis of the dynamic and fine-grained sparsity structure in diffusion attention was viewed as well motivated and informative, and the proposed method was considered coherent, modular, and easy to integrate across backbones. Reviewers also found the empirical results support a favorable trade-off between efficiency and generation quality, and the rebuttal resolved most of the major technical questions. Remaining concerns primarily center on originality relative to prior works. Overall, the paper represents a solid contribution, placing it above the acceptance threshold.